# Are medical outliers associated with worse patient outcomes? A retrospective study within a regional NHS hospital using routine data

Neophytos Stylianou,[1,2] Robin Fackrell,[2] Christos Vasilakis[1]

► Prepublication history and additional material is available. To view please visit the journal (http://dx.doi.org/ 10.1136/bmjopen-2016-015676).

## ABSTRACT

**Objective** To explore the quality and safety of patients' healthcare provision by identifying whether being a medical outlier is associated with worse patient outcomes. A medical outlier is a hospital inpatient who is classified as a medical patient for an episode within a spell of care and has at least one non-medical ward placement within that spell.

**Data sources** Secondary data from the Patient Administration System of a district general hospital were provided for the financial years 2013/2014–2015/2016. The data included 71 038 medical patient spells for the 3-year period.

**Study design** This research was based on a retrospective, cross-sectional observational study design. Multivariate logistic regression and zero-truncated negative binomial regression were used to explore patient outcomes (in-hospital mortality, 30-day mortality, readmissions and length of stay (LOS)) while adjusting for several confounding factors.

**Principal findings** Univariate analysis indicated that an outlying medical in-hospital patient has higher odds for readmission, double the odds of staying longer in the hospital but no significant difference in the odds of in-hospital and 30-day mortality. Multivariable analysis indicates that being a medical outlier does not affect mortality outcomes or readmission, but it does prolong LOS in the hospital.

**Conclusions** After adjusting for other factors, medical outliers are associated with an increased LOS while mortality or readmissions are not worse than patients treated in appropriate specialty wards. This is in line with existing but limited literature that such patients experience worse patient outcomes. Hospitals may need to revisit their policies regarding outlying patients as increased LOS is associated with an increased likelihood of harm events, worse quality of care and increased healthcare costs.

## Strengths and limitations of this study

► There is scarce evidence associating medical outliers and patient outcomes.
► This is the first quantitative study in England's NHS that has sought to investigate potential associations between medical outliers and patient outcomes.
► Adjusting for several patient-specific confounding factors indicates that medical outliers are only associated with longer length of stay.
► Large data set used increases the power of the study and minimises the single-centre limitation.
► Routinely collected data limit the association adjustment process, but on the other hand make this study feasible and efficient.

¹Centre for Healthcare Innovation & Improvement (CHI²), School of Management, University of Bath, Bath, UK
²Royal United Hospitals Bath NHS Foundation Trust, Bath, UK

**Correspondence to**
Dr. Neophytos Stylianou;
n.stylianou@bath.ac.uk and
Prof. Christos Vasilakis;
c.vasilakis@bath.ac.uk

## INTRODUCTION

Faced with ever-increasing pressures and targets, hospital bed managers often resort to placing patients on wards that are not specifically designed or designated for the type of care patients require. Although patients may be admitted to a ward faster and beds across the entire hospital are used more, patients may not receive treatment in the way it was designed for or by the nurses that are specialised in their care. This phenomenon is commonly known as 'outliers', but it can also be referred to in the literature as 'outlying hospital in-patients', 'overflow', 'sleep-outs' or 'boarders'.[1 2]

Hospital wards are broadly categorised as medical or surgical, with each category having a number of generic and specialised wards hosting the delivery of care for patients with similar diagnoses.[3] Examples include patients with stroke being cared for in an acute stroke ward and patients who underwent open heart surgery in a cardiac surgery ward. The goal is that by clustering clinical and nursing skills and specialised equipment in a single point in the hospital, that is, the specialist ward, the clinical management of patients improves and provides an environment that is purposefully designed to meet the clinical and non-clinical needs of patients.[4] By placing patients on a ward designed to treat their condition, the expectation is that their clinical needs are better met.[3]

Nursing and clinical staff providing care for outlying patients are faced with several challenges. The care provided may not be the most appropriate or as timely as it could be, as patients are placed on wards with staff that do not have specific expertise for the patients' condition.[1 2] Some empirical evidence on medical outliers has suggested that patients who are chosen to be medical outliers are more medically 'fit'.[5–8] Implicitly, patients who are classified as 'fit' to be outliers may be perceived by staff to be of lower priority.[9] The presence of medical outliers may mean longer ward rounds for the clinical teams and thus contributing to valuable clinical time being wasted.[10] Another empirical study in an Australian hospital showed that medical outliers have a higher frequency of emergency calls leading to higher workload on staff who do not know the patients as well, leading to suboptimal decisions that might, in turn, negatively impact on safety.[9]

This placement of patients in care wards that do not offer the specialised care that patients need may lead to a suboptimal and fractured provision of care. Continuity of care, a fundamental aspect of high-quality care, is based on establishing relationships with ward staff (continuity of relationship) and their medical management (continuity of management).[11 12] A recent report that looked at continuity of care for elderly patients showed that patients are frequently moved around from bed to bed within a ward and to different wards.[11] These movements, often happening out-of-hours and without the patient being informed of the reasons, have shown to be potentially unpleasant and stressful , have a detrimental effect on patient experience and compromise continuity of care.[11]

The phenomenon of medical outliers is more common in publicly funded health systems.[13] It has been observed in England, Wales, Spain, France, Italy, Sweden, Australia and New Zealand, but as it has not been extensively researched there is scarce good-quality evidence available.[1 2 9 13–21] In England's NHS it is an issue often documented in internal hospital reports, with some organisations having put in place standard operating procedures and process pathways indicating when a patient can be considered an outlier and what process to be followed.[6–8 22 23] In recent years, two qualitative studies on medical outliers in the NHS highlighted the problem and raised the level of concern of NHS staff regarding patient safety.[1 2] The lack of quantitative studies in this area may be due to the fact that the relevant information is not routinely recorded or collected by hospital information systems or national hospital databases.

Thus, our aim was to investigate potential associations between medical outliers and patient outcomes using routinely collected hospital data. Our working hypothesis is that outlier patients are associated with worse patient outcomes. To the best of our knowledge, this is the first quantitative study in the UK that investigated the effects medical outliers have on patient outcomes regardless of diagnosis.

## METHODS

Data for this retrospective observational study were provided by the Business Intelligence Unit of the Royal United Hospitals Bath NHS Foundation Trust, a district general hospital in South West England that serves a population of 550 000 people with approximately 565 beds. The time period of the study covered three financial years, 2013/2014–2015/2016. Because this study used anonymised, non-identifiable secondary data, an ethics approval was not necessary according to the NHS regulations.[24] A medical outlier was defined as a hospital inpatient who was classified as a medical patient for an episode within a spell of care and had at least one ward placement on a non-medical ward within that spell. An episode of care was the time a patient spent under the care of one consultant. Many episodes comprised a spell of care that was the continuous, usually uninterrupted, stay of a patient within a hospital provider.[25]

The data were provided in a hierarchical structure as follows: (1) patient ward moves within episodes of care, (2) patient episodes of care and (3) patient spells. We used this data structure to establish whether a patient had been classified as an outlier at any point within his/her spell of care at the hospital. We then used the dominant episode data for further analysis. A dominant episode is the first episode in a multiepisode spell that contains the primary procedure, and if no primary procedure exists then it is the first episode that contains the primary

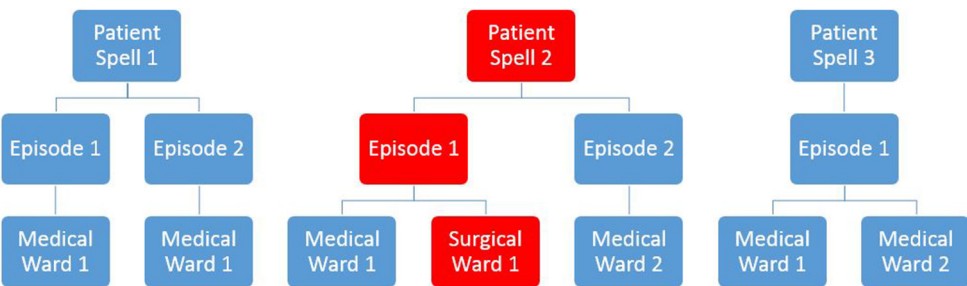

**Figure 1** An illustrative example of data structure and classification scheme used to define a medical outlier (Patient Spell 2 in this example). Only medical spells were considered, which are the spells with the dominant episode being allocated to a medical specialty.

diagnosis for the spell.[26] A schematic illustration of the data set structure is shown in figure 1.

The patient outcomes investigated were in-hospital mortality, 30-day mortality (death within 30 days of discharge), readmission to the same hospital within 30 days of discharge and hospital length of stay (LOS). For the patient outcome of readmission, patients who died in the hospital were excluded.

Univariate analysis was used to establish any relationship between the patient outcome under investigation when the patient was either an outlier or not. Logistic regression was used for readmission and mortality outcomes, and zero-truncated negative binomial regression was used for LOS examination. The multivariable methods were used to adjust for potential confounding factors that are already known to be associated with the outcomes under investigation. Confounding factors that were adjusted for were age, sex, primary diagnosis using the International Statistical Classification of Diseases and Related Health Problems 10th revision (ICD-10) category (of the dominant episode), admission method (emergency or not), count of procedures, count of 'secondary' diagnosis, weekend admission, weekend discharge and socioeconomic status of the patient. Socioeconomic status was measured using the 2015 Index of Multiple Deprivation (IMD) score at Lower Super Output Areas (LSOA). IMD is a measure of deprivation calculated using seven domains of deprivation.[27] LSOAs are small geographical areas of roughly equal population size, averaging 1500 residents or 650 households.[28] The smaller the score the more deprived an area is. All statistical significance was measured at a p value of less than 0.05. All analyses were performed using Stata V.13.1.[29]

## RESULTS
### General description of the data
A full extract of the Patient Administration System of the hospital was provided for the financial years 2013/2014–2015/2016. This extract was made up of 71 038 dominant spells in which patients were classified as medical cases (as opposed to surgical or maternity). A general description of the data is shown in table 1. Age, sex, secondary diagnoses and procedure counts did not present significant differences. The mean (SD) age of patients admitted to the hospital was 66.40 (20.66), and the age distribution for medical outliers and non-outliers was similar. Male spells accounted for 47.61% of the total admitted spells and the remaining 52.39% were women. In-hospital mortality was roughly 5% of all spells for the 3 years of the data and 30-day mortality was 3.26%. Of the 71 038 spells of care, 9.88% have been outlying at some point of their care. Approximately 23% of the spells were weekend admissions, with 14.10% outlying. IMD ranged from 32 to 32 812 and the mean (SD) was 20 342.4 (7991.13). Medical outliers and non-outliers spells had relatively similar IMD scores.

On average, patient spells had six secondary diagnoses attached to their medical record. Non-outliers had 6.17 and medical outliers had 6.85, a difference that was statistically significant (p<0.001). The same applied for procedure numbers, but those were on average three. Regarding the overall primary diagnosis, the most common was signs and symptoms that do not give a definitive diagnosis. This was followed closely by diseases of the circulatory system. The primary diagnosis was different between non-medical outliers and medical outliers. For non-medical outliers, the most common primary diagnosis was diseases of the circulatory system, followed closely by category symptoms and signs and abnormal clinical findings not elsewhere classified, whereas for medical outliers the most common primary diagnosis category was diseases of the respiratory system, followed by diseases of the digestive system.

### Medical outliers and patient outcomes
Table 2 presents the patient outcomes for both medical and non-medical outliers. There were 3531 in-hospital deaths, of which 344 (9.74%) was attributed to medical outliers. A similar percentage (10.50%) was observed for 30-day mortality. Approximately 11% of the total readmissions to the hospital was to patients who were medical outliers at some point during their care. Univariate analysis of medical outliers on the patient outcome revealed that medical outliers are not associated with in-hospital or 30-day mortality, but they do affect the readmission probabilities with statistically significant OR (95% CI) of 1.193 (1.110 to 1.282).

Table 2 shows that outliers have more than double the LOS of non-medical outlier spells. As the distribution of LOS is highly positively skewed, median (IQR) is reported. LOS is further described in figure 2, where the distribution of both spell groups is shown. It is clear that medical outliers are staying in the hospital for longer periods of time and more patient spells have recorded more than 30 days in the hospital compared with non-medical outliers. Univariate analysis for LOS indicated that the log count for medical outlier patients is 0.672 more than non-medical outliers, equating to staying in hospital approximately 2 days longer than non-medical outliers.

After adjusting for variables that may affect the patient outcomes we were investigating, it was found that medical outliers are not associated with increased odds of in-hospital mortality, 30-day mortality or readmission (table 3). However, medical outliers were found to be associated with an increasing effect on LOS. Medical outliers compared with non-medical outliers, while adjusting for all other variables and keeping them constant, have a higher log count of stay of 0.042. The expected LOS for non-medical outliers was 3.86 (95% CI 3.22 to 4.50), while for medical outliers was 4.03 (95% CI 3.36 to 4.7). In other words, medical outliers spent 0.17 days (p value: 0.006) more in hospital compared with non-medical outliers.

**Table 1** General description of the data

| Characteristics | Non-outlying | Outlying | p Value | Total |
|---|---|---|---|---|
| Spells | 64 017 (90.12) | 7021 (9.88) | – | 71 038 |
| Gender | | | | |
| Male | 30 487 (90.14) | 3333 (9.86) | 0.809 | 33 820 |
| Female | 33 530 (90.09) | 3688 (9.91) | | 37 218 |
| Age* | 66.16 (20.64) | 68.62 (20.62) | <0.001 | 66.40 (20.66) |
| Secondary diagnosis numbers* | 6.17 (3.74) | 6.85 (3.83) | <0.001 | 6.24 (3.76) |
| Procedure numbers* | 1.24 (2.16) | 1.81 (2.81) | <0.001 | 3.29 (2.49) |
| Weekend admissions | 13 861 (85.90) | 2276 (14.10) | | 16 137 |
| IMD* | 20 360.08 (7992.15) | 20 181.38 (7980.63) | 0.079 | 20 342.4 (7991.13) |
| Primary diagnosis ICD-10 category | | | | |
| Certain infectious and parasitic diseases | 2470 (3.86) | 352 (5.01) | – | 2822 (3.97) |
| Neoplasms | 2460 (3.84) | 381 (5.43) | | 2841 (4.00) |
| Diseases of the blood and blood-forming organs and certain disorders involving the immune mechanism | 760 (1.19) | 68 (0.97) | | 8289 (1.17) |
| Endocrine, nutritional and metabolic diseases | 1521 (2.38) | 146 (2.08) | | 1667 (2.35) |
| Mental and behavioural disorders | 1426 (2.23) | 117 (1.67) | | 1543 (2.17) |
| Diseases of the nervous system | 2236 (3.49) | 118 (1.68) | | 2354 (3.31) |
| Diseases of the eye and adnexa | 137 (0.21) | 11 (0.16) | | 148 (0.21) |
| Diseases of the ear and mastoid process | 206 (0.32) | 32 (0.46) | | 238 (0.34) |
| Diseases of the circulatory system | 12 283 (19.19) | 522 (7.43) | | 12 805 (18.03) |
| Diseases of the respiratory system | 9139 (14.28) | 1277 (18.19) | | 10 416 (14.66) |
| Diseases of the digestive system | 2934 (4.58) | 1022 (14.56) | | 3956 (5.57) |
| Diseases of the skin and subcutaneous tissue | 1013 (1.58) | 276 (3.93) | | 1289 (1.81) |
| Diseases of the musculoskeletal system and connective tissue | 2058 (3.21) | 351 (5.00) | | 2409 (3.39) |
| Diseases of the genitourinary system | 3019 (4.72) | 671 (9.56) | | 3690 (5.19) |
| Pregnancy, childbirth and the puerperium | 109 (0.17) | 29 (0.41) | | 138 (0.19) |

Continued

**Table 1** Continued

| Characteristics | Non-outlying | Outlying | p Value | Total |
|---|---|---|---|---|
| Congenital malformations, deformations and chromosomal abnormalities | 19 (0.03) | 2 (0.03) | | 21 (0.03) |
| Symptoms, signs and abnormal clinical and laboratory findings, not elsewhere classified | 12 026 (18.79) | 852 (12.14) | | 12 878 (18.13) |
| Injury, poisoning and certain other consequences of external causes | 9970 (15.57) | 777 (11.07) | | 10 747 (15.13) |
| Factors influencing health status and contact with health services | 215 (0.34) | 16 (0.23) | | 231 (0.33) |
| Unknown | 16 (0.02) | 1 (0.01) | | 17 (0.02) |

All values are numbers (percentages) unless otherwise stated.

t-Tests and $\chi^2$ tests were used to compare continuous and categorical variables, respectively.

*Mean (SD).

ICD-10, International Statistical Classification of Diseases and Related Health Problems 10th revision; IMD, Index of Multiple Deprivation.

## DISCUSSION

Our analysis showed that approximately 10% of the patients classified as medical patients were outliers at least once during their entire hospitalisation. The results indicated that age, comorbidity, emergency admissions within the same year, primary diagnosis category and weekend admission impose significant effects on in-hospital mortality, 30-day mortality, readmissions and LOS. These factors are known to affect the patient outcomes we used in this study. Being a medical outlier does not appear to affect mortality, either in-hospital or 30 days after discharge. This was also observed after controlling for confounding variables. The preventive odds observed for mortality outcomes during the multivariable analysis can be explained by the fact that patients classified based on medical expertise as being more severely affected by their disease/condition are kept on wards designated and specific for their condition, and patients who become outliers are typically less unwell.

Being a medical outlier is associated with higher odds of being readmitted to the same hospital within 30 days, but after adjusting for a number of confounding variables this association was diluted. However, we found that outlier patients have double the LOS of non-outlying patients, and multivariable analysis confirmed this increase in LOS.

Our results are in line with those found in the extant literature comprising a limited number of studies. Specifically, Alameda and Suárez in a 2009 study performed in Spain found that readmissions were not affected by being a medical outlier but that there was an increasing effect on LOS.[13] Longer LOS for outlying patients was also observed by Stowell *et al* in a French study in 2013.[18]

The increased LOS medical outliers have compared with non-medical outliers could be explained by the difference in primary disease diagnosis. As stated

**Table 2** Patient outcomes and medical outliers

| Characteristics | Non-outlying | Outlying | Effect size (95% CI)* | p Value |
|---|---|---|---|---|
| In-hospital mortality | 3187 (90.26) | 344 (9.74) | 0.983 (0.877 to 1.102) | 0.773 |
| 30-day mortality | 2072 (89.50) | 243 (10.50) | 1.071 (0.936 to 1.227) | 0.315 |
| Readmission | 7592 (88.66) | 971 (11.34) | 1.193 (1.110 to 1.282) | <0.001 |
| LOS† | 3 (7) | 7 (7) | 0.672 (0.624 to 0.719) | <0.001 |

All values are numbers (percentages) unless otherwise stated.

*OR with the exception of LOS where the negative binomial regression coefficient is given.

†Median (IQR).

LOS, length of stay.

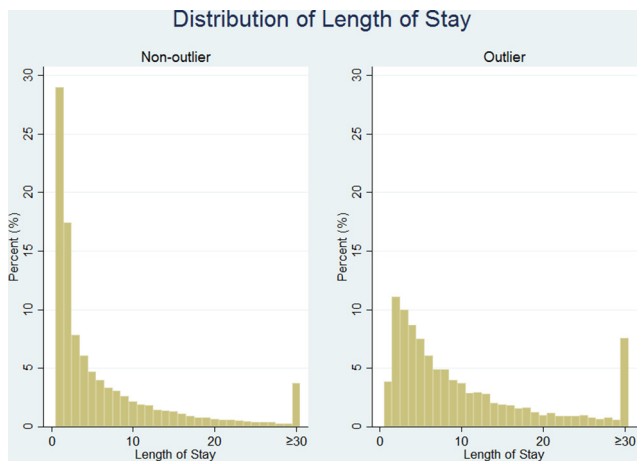

**Figure 2** Distribution of length of stay for medical outlier and non-medical outliers.

previously the primary diagnosis differed between the two groups, with outliers being predominantly diagnosed with circulatory and respiratory diseases, whereas non-outliers were primarily diagnosed with circulatory diseases and symptoms that could not be classified. LOS is an important metric of quality of healthcare provision as it is often associated with complications occurring within the hospital.[30–32] Further reasons why medical outlying patients may experience longer LOS include the delayed medical contact between arrival and first examination in the case the patient was placed straight away on an 'inappropriate' ward, insufficient contact with medical and nursing personnel during outlying period, and also inability of nursing personnel to meet patient needs because of lack of specialisation on the disease/condition of the patient.

The field of medical outliers is not extensively researched, although evidence does exist to suggest that being in the right bed at the right time is beneficial to patients, as a number of studies in stroke, burns, asthma and gastrointestinal haemorrhage have shown.[33–36] However, there is not enough evidence in the literature to give a definitive answer on the effect of medical outliers on patient outcomes. This is because some studies were only looking at qualitative data to understand what causes medical outliers, or because the

**Table 3** Effect sizes of medical outliers on patient outcomes adjusting for other independent variables in the multivariable models

| Patient outcome | Effect size* | Significance (0.05) | (95% CI) |
|---|---|---|---|
| In-hospital mortality | 0.679 | <0.001 | 0.600 to 0.769 |
| 30-day mortality | 0.711 | <0.001 | 0.614 to 0.822 |
| Readmission | 0.928 | 0.09 | 0.849 to 1.013 |
| Length of stay | 0.042 | 0.004 | 0.014 to 0.071 |

*OR with the exception of LOS where the zero-truncated negative binomial coefficient is given.

quantitative studies were based on small sample sizes with a lot of exclusion criteria or performed in a single hospital department or ward. Our study is the first to analyse the effect medical outliers have on patient outcomes using a large set of routinely collected data not limited to a specific clinical service in the UK.

### Limitations
Limitations are inevitable in retrospective, observational studies. Our study used routinely collected data from the hospital's Patient Administration System, which is known to be a limiting factor in studies. The secondary data usage is of limited use for confounding factors as we could not adjust for factors we believed would be important in the multivariable models we developed. For example we could not adjust for the severity or comorbidity of a patient, although for the latter we used the proxies of the number of procedures and diagnosis codes recorded in the patient record for that spell. A prospective study could allow for the collection of specific clinical data such as severity of the patient and other factors associated with patient movement, and outlier status such as time to examine patients on ward rounds or patient complications, but this would need a lot of resources to be able to get as many data.

Our study is a single-centre study that in theory minimises generalisability, although the amount of routinely collected data we used somewhat mitigates this limitation. Given the lack of relevant data being recorded in national data sets such as Hospital Episode Statistics, only a study that includes prospective data collection method could overcome this limitation fully.

Reasons for longer patient LOS could not be investigated as they are not typically recorded in administrative databases. We could not identify nursing and medical staff numbers per ward, experience of nurses and doctors, delay in prescription of drugs or laboratory analyses, and mistakes occurring during prescription or diagnosis. These could be further investigated to elucidate what makes outlying patients more susceptible to different patient outcomes compared with non-outlying patients.

Longer LOS in hospital makes the patients susceptible to hospital-acquired infections and could be further investigated. Again, infection data are not routinely collected for every patient admission.

Finally, our study investigated medical outliers only and not surgical or maternity cases. A recent publication by the Royal College of Surgeons of England showed that surgical patients also experience outlying, which raises the same concerns regarding patient safety, continuity of care and overall levels of provision of high-quality healthcare.[10]

### CONCLUSIONS
There is plenty of anecdotal and some research evidence to suggest that, in busy acute care hospitals, the phenomenon of medical outliers is commonplace. In the district general

hospital that we studied using routinely collected data over a 3-year period, we found that almost one in 10 medical patient spells spent at least part of their stay in a non-medical ward.

After adjusting for other factors, we found medical outliers were associated with an increased LOS. We also found that mortality or readmissions were not worse than patients treated in appropriate specialty wards. This is in line with existing but limited literature that such patients experience worse patient outcomes. However, unlike previous studies on this issue, our study was not limited to a single clinical service.

In light of our findings, hospitals may need to revisit their policies regarding outlying patients as there is evidence suggesting that longer LOS is associated with an increased likelihood of harm events, worse quality of care and increased healthcare costs.

However, on the balance of evidence, we cannot advise on taking steps towards eliminating the practice entirely as that may have negative implications on bed availability, which in turn may lead to increases in cancellations of elective cases and longer emergency department waiting times for hospital admission among other negative consequences.

Rather, our findings could be put in better use by informing discussions between clinicians and managers at hospital, specialty and ward levels of the negative effects associated with medical outliers. These effects should be taken into account when devising policies and procedures (eg, in ring-fencing pools of beds for specific types of patients), or when deciding on bed management issues on an ad hoc basis.

As it is common with secondary data analysis studies, more research is needed to investigate the effects more thoroughly through, for example, a multicentre study of prospectively collected data. The call for additional research is particularly relevant here as the practice of placing patients on any available hospital bed appears to be widespread.

**Contributors** NS, RF and CV conceived the idea for the study. NS and CV designed the study. NS analysed all data and wrote the original manuscript in conjunction with CV. RF facilitated the acquisition of data and provided clinical expert guidance. All authors commented on the manuscript.

**Funding** The Researchers-in-Residence programme of research was funded by the Royal United Hospitals NHS Foundation Trust.

**Competing interests** We would like to declare that Robin Fackrell is the Head of Medical Division at the RUH Bath. Christos Vasilakis and Neophytos Stylianou both received funding form Royal United Hospitals of Bath NHS Trust to perform this research as part of the Researchers-in-Residence programme.

**Provenance and peer review** Not commissioned; externally peer reviewed.

**Data sharing statement** Data were provided by the Business Intelligence Unit of Royal United Hospitals of Bath. As we cannot keep the data based on data sharing agreements in place, data can be requested by the provider if data sharing agreement requirements are met by recipient.

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
