## [Reviewer comments · BMJ Open]

ARTICLE DETAILS

TITLE (PROVISIONAL)	Are medical outliers associated with worse patient outcomes? A retrospective study within a regional NHS hospital using routine data
AUTHORS	Stylianou, Neophytos; Fackrell, Robin; Vasilakis, C

VERSION 1 - REVIEW

REVIEWER	Dr Lucy Goulding King's College London - UK
REVIEW RETURNED	20-Jan-2017

GENERAL COMMENTS	Thank you very much for the opportunity to review this interesting paper that highlights a very important quality and safety issue. I'd like to bring my PhD thesis to the attention of the authors. This is published online at: http://etheses.whiterose.ac.uk/2400/ and is therefore possible to reference. The authors will see that I used a very similar methodology within my own analyses of outliers using routinely collected data. My findings were similar (e.g. outliers had a significantly longer length of hospital stay adjusting for other factors but there was no significant difference in mortality). It would in my opinion be helpful to make reference to my study as the findings of the two studies support one another. This is particularly important as both studies were conducted in a single Trust in England, therefore alone the findings lack generalisability. I'd also therefore make the point that this is not the first study in the UK of this type (please amend throughout e.g. page 5 line 27). My opinion is that the manuscript would also benefit from the following revisions: ABSTRACT: - Adjusting for 'risk factors' – do you mean confounding factors?- Typo in the principle findings section requires correction - "but not significant difference".- The abbreviation 'LOS' is used in the abstract without being written out in full.- The written English within the strengths and limitations section of the abstract should be improved INTRODUCTION: - Page 4 Lines 34 and 38 – you could reference my qualitative studies ([1] and [2] in your reference list) here?- Page 4 Line 34 is not currently backed up by a reference.- The paragraph on continuity of care within the introduction (page 4) could be written more succinctly.- Line 48 "inappropriate care wards" – not sure this sounds quite right
--

	- Page 5 line 7 – please provide a reference (to support the statement that outlying is more common in publicly funded systems) METHODS: - Page 5 lines 49 – 56 – the description of the methods is not clear to me. - Page 6 lines 54 – 58 – I’m not sure what this means, please improve the clarity of the description of the methods. RESULTS: - Please improve the labelling and clarity of Table 1. By ‘total’ do you mean outliers and non-outliers combined? Is this useful information to present? - The written English in the ‘Medical Outliers and patients outcomes’ section requires improvement. - With regards the length of stay findings, it is not possible to assume causality. E.g. people may be more likely to stay in hospital for longer because they are an outlier OR staying in hospital for a long time means that there is more opportunity to become an outlier at some point. This could also be better addressed in the discussion. - More could be done to make the language within the results section accessible to readers who are less familiar with the statistical methods adopted. DISCUSSION: - Page 10, lines 31-39 – please provide references to back up your hypotheses. - The written English in the conclusion requires improvement. FIGURE: - The presentation of Figure 1 could be improved. ‘Length’ is written incorrectly several times.
--	---

REVIEWER	Lua Perimal-Lewis Flinders Digital Health Research Centre School of Health Sciences Faculty of Medicine, Nursing and Health Sciences Flinders University GPO Box 2100 Adelaide SA 5001 Australia Tonsley Campus 1284 South Road Clovelly Park SA 5042 Australia
REVIEW RETURNED	24-Jan-2017

GENERAL COMMENTS	This manuscript analysed a large NHS dataset to investigate the outcomes of care for medical outliers showing that outliers’ LOS is longer than the non-outlying group. The following recommendations are important to be carried out to improve the quality of statistical analysis and writing. 1. Introduction – improve the definition/introduction/background of outliers to include other available literature on this topic. This phenomenon is also experienced in other countries and there are literature to support this. Literature review should be improved to include other similar findings on outliers outside of NHS. Two additional references to outlying phenomenon in Australia:  • Relationship between in-hospital location and outcomes of care in patients of a large general medical service - L. Perimal-Lewis, J. Y. Li, P. H. Hakendorf, D. I. Ben-Tovim, S. Qin and C. H. Thompson
--

• The relationship between in-hospital location and outcomes of care in patients diagnosed with dementia and/or delirium diagnoses: analysis of patient journey. - Perimal-Lewis L, Bradley C, Hakendorf PH, Whitehead C, Heuzenroeder L, Crotty M

2. Episode of care vs spell of care – I think these phrases means the same. If not please define/clarify their differences. To reduce confusion it might be better to use one phrase or clearly define.

3. Lines 49 –52 on page 5: would it be possible to provide a snapshot of example data to show/differentiate between i), ii) and iii)?

4. Lines 17 - 21 on page 6: list the potential admission source and discharge method. Why admission source, discharge method and socioeconomic status were used as confounding factors? What is the reasoning behind choosing these variables and how it relates to outliers should be discussed in the discussion section.

5. Line 56 on page 6 – ...”both outlier groups”?

6. Method section – in the statistical analysis paragraph; please indicate when a p-value is considered statistically significant. Explain the purpose of using univariate and multivariate. Some results mention univariate and some multivariate; this is rather confusing. Present all univariate results in one table including their p-values and a separate table/s for multivariate results including p-values. The table/s should show values for each confounder. What is the reason for using negative binomial regression?

7. Table 1 on page 7 – please provide the p-value comparing ‘non-outlying ‘and ‘out-lying’ group.

8. Table 2 on page 8 – please provide the p-value comparing ‘non-outlying ‘and ‘out-lying’ group.

9. For Figure 1 – address in discussion what might be the reason for the big variation in 1 day LOS.

10. Lines 11 – 25 on page 9; ...”keeping other variables constant” or a variation of this phrase is repetitive. Explain at the start about the characteristics of multivariate modelling once and refrain from repeating.

11. Table 3 on page 9 – could be deleted if the suggested tables above are completed.

12. Line 3 on page 10 – first time usage of ‘home ward’ without prior definition. Please define.

13. Line 8, page 10 – what does it mean by ‘partial’ adjusting?

14. Lines 51-54, page 10 – Disagree with this statement; both the papers above also uses large routinely collected data sets.

15. Lines 10-13 on page 11 – What makes number of procedures and diagnosis codes proxies for severity? Severity tells how urgently a patient has to be attended in the Emergency Department based of the presenting condition?

16. Further clarifications needed on:

a. It would be good to understand the percentage of time spent in outlier ward/s and percentage of time spent in home ward/s for both groups.

b. Statistical analysis needs further work. For example presenting univariate results when multivariate modelling was done does not add value but is confusing.

c. When analysing readmission – were the patients who died in-hospital removed from the analysis?

17. There are minor spelling and grammatical errors, which need attention.

VERSION 1 – AUTHOR RESPONSE

Reviewer 1 Lucy Goulding

ABSTRACT:

1 Adjusting for 'risk factors' – do you mean confounding factors?

We are using the term 'confounding factor' now throughout.

2 Typo in the principle findings section requires correction - "but not significant difference".

Typo corrected

3 The abbreviation 'LOS' is used in the abstract without being written out in full.

Changed

4 The written English within the strengths and limitations section of the abstract should be improved

That section has been made more concise

Introduction

5 Page 4 Lines 34 and 38 – you could reference my qualitative studies ([1] and [2] in your reference list) here?

We have added these two citations.

6 Page 4 Line 34 is not currently backed up by a reference.

We have used one of the references from the previous comment (5) to back up the assertion

7 The paragraph on continuity of care within the introduction (page 4) could be written more succinctly.

We have redrafted the paragraph for clarity and succinctness.

8 Line 48 "inappropriate care wards" – not sure this sounds quite right

We agree inappropriate is a bit heavy so we changed it to:

"care wards which do not offer the specialised care the patient needs"

9 Page 5 line 7 – please provide a reference (to support the statement that outlying is more common in publicly funded systems)

Thanks for pointing it out. Reference now added

Methods

10 Page 5 lines 49 – 56 – the description of the methods is not clear to me.

We have made changes that hopefully improve the textual description of the methods and added a figure to better describe the data structure and classification of a medical outlier.

11 Page 6 lines 54 – 58 – I'm not sure what this means, please improve the clarity of the description of the methods.

Thank you for pointing this out. The second reviewer also made a similar comment. We meant to say that both outliers and non-outlying patients had seven secondary diagnosis in their record. This has now been addressed.

Results

12 Please improve the labelling and clarity of Table 1. By 'total' do you mean outliers and non-outliers combined? Is this useful information to present?

Yes total means all patients. As with every table used in quantitative studies the total is always presented although it is just a simple addition of the two groups, but it is used in order to make results easily visible to the reader.

13 The written English in the 'Medical Outliers and patients outcomes' section requires improvement
This section has now been improved.

14 With regards the length of stay findings, it is not possible to assume causality. E.g. people may be more likely to stay in hospital for longer because they are an outlier OR staying in hospital for a long time means that there is more opportunity to become an outlier at some point. This could also be better addressed in the discussion.

We agree that our study was not designed to identify causation. What we state in the discussion is that being an outlier is associated with longer LOS and we also gave possible explanations about this, trying to avoid causal inference.

15 More could be done to make the language within the results section accessible to readers who are less familiar with the statistical methods adopted.

We tried to keep the language as non-technical as possible. When we mention more technical terms we are trying to also provide a more "lay" explanation as well.

Discussion

16 Page 10, lines 31-39 – please provide references to back up your hypotheses.

These are simply possible explanations not supported by literature necessarily. We have revised the text to make this clear.

17 The written English in the conclusion requires improvement.

We have made revisions that we hope have improved that section of the paper.

Figure

18 The presentation of Figure 1 could be improved. 'Length' is written incorrectly several times.

Thanks for spotting this error. "Length" has now been corrected.

Reviewer 2 Lua Perimal Lewis

1 Introduction – improve the definition/introduction/background of outliers to include other available literature on this topic. This phenomenon is also experienced in other countries and there are literature to support this. Literature review should be improved to include other similar findings on outliers outside of NHS. Two additional references to outlying phenomenon in Australia:

- Relationship between in-hospital location and outcomes of care in patients of a large general medical service - L. Perimal-Lewis, J. Y. Li, P. H. Hakendorf, D. I. Ben-Tovim, S. Qin and C. H. Thompson
- The relationship between in-hospital location and outcomes of care in patients diagnosed with dementia and/or delirium diagnoses: analysis of patient journey. - Perimal-Lewis L, Bradley C, Hakendorf PH, Whitehead C, Heuzenroeder L, Crotty M

Thanks for sharing those two papers with us. We could not have known of the second one as it was published after we finished the research and were in the process of submitting the paper.

We believe we did include other available literature on the topic and we did not limit it to the England's NHS. We mention in the introduction a study performed in Australia specifically. We discuss the literature further in our discussion where we mention more studies of outliers effect on outcomes in various countries such as Italy, Spain, France.

Both suggested papers are now included in our paper.

2 Episode of care vs spell of care – I think these phrases means the same. If not please define/clarify their differences. To reduce confusion it might be better to use one phrase or clearly define.

Actually those phrases mean different things that are defined at the end of the first paragraph of the Methods section. A reference was also given for further information.

3 Lines 49 –52 on page 5: would it be possible to provide a snapshot of example data to show/differentiate between i), ii) and iii)?

We have made changes that hopefully improve the textual description of the methods and added a figure to better describe the data structure and classification of a medical outlier.

4 Lines 17 - 21 on page 6: list the potential admission source and discharge method. Why admission source, discharge method and socioeconomic status were used as confounding factors? What is the reasoning behind choosing these variables and how it relates to outliers should be discussed in the discussion section.

Those are known confounding factors that affect the outcomes we chose to investigate. In the final models that adjust for confounding factors admission source and discharge destination is not used as it was proven not to be associated with the outcomes under investigation although in the literature there are evidence suggesting that. To prevent any confusion we removed them from the list of confounding factors in the methodology section.

We made it clear now in the methods section that the confounding factors which were adjusted for, are ones that are known to be associated with the outcomes under investigation.

5 Line 56 on page 6 – ...”both outlier groups”?

The text has now been made clear.

“On average, patient spells had six secondary diagnoses attached to their medical record. Non outliers had 6.17 and medical outliers had 6.85, a difference which was statistically significant ($p < 0.001$).

6 Method section – in the statistical analysis paragraph; please indicate when a p-value is considered statistically significant. Explain the purpose of using univariate and multivariate. Some results mention univariate and some multivariate; this is rather confusing. Present all univariate results in one table including their p-values and a separate table/s for multivariate results including p-values. The table/s should show values for each confounder. What is the reason for using negative binomial regression? Significance level now added to the methods section.

We are now describing the use of univariate and multivariate methods in the methodology section. In the results section we believe we followed a logical order. We present the univariate results, describe them and then move on to the multivariate findings.

Unfortunately presenting all the multivariable models in the paper would have made the paper heavy with tables (which would not add much) thus we decided to just present the effect size for medical outliers. Some of the variables used are categorical variables with many categories, such as primary diagnosis group having 21 categories. For further justification why we gave the univariate and multivariable please see comment 16b.

LOS is measured in days so we considered LOS as a count variable. As we mentioned we only included in our analysis inpatient data which means that they were admitted. We considered admission as occupying a bed thus the patients had at least a LOS of 1 day. This is what drove us to use the zero truncated Negative Binomial regression. A zero truncated negative binomial regression was used as the LOS was a count variable and not a binary one to use logistic regression as we did with the other three outcome variables under investigation. We chose the negative binomial regression over the Poisson regression as our data were found to be over dispersed. This overdispersion is an indication that they are better estimated using negative binomial model than a Poisson model. The overdispersion was tested using the dispersion parameter being significantly greater than zero. If it was equal to zero then the model would be reduced to a simple Poisson model.

7 Table 1 on page 7 – please provide the p-value comparing ‘non-outlying’ and ‘out-lying’ group. P-values now added where appropriate

8 Table 2 on page 8 – please provide the p-value comparing ‘non-outlying’ and ‘out-lying’ group.
P-values were added in table 2

9 For Figure 1 – address in discussion what might be the reason for the big variation in 1 day LOS.
Several reasons are already given in discussion section on why outliers may have higher LOS.

10 Lines 11 – 25 on page 9; ...”keeping other variables constant” or a variation of this phrase is repetitive. Explain at the start about the characteristics of multivariate modelling once and refrain from repeating.

Thanks for pointing it out. This has now been addressed in the paper.

11 Table 3 on page 9 – could be deleted if the suggested tables above are completed.
As responded for comment number 16 we agree that when doing multivariable analysis is better if you include the full multivariable model. Unfortunately in this instance we have four multivariable models with many categorical variables in each one so that would take up a lot of space in the article and would not have added much value. If you still believe we should include them we can provide them as supplementary information.

12 Line 3 on page 10 – first time usage of ‘home ward’ without prior definition. Please define.
We have replaced the term “home ward” with
“wards designated and specific for their condition”.

13 Line 8, page 10 – what does it mean by ‘partial’ adjusting?
The term “partial” has now been removed.

14 Lines 51-54, page 10 – Disagree with this statement; both the papers above also uses large routinely collected data sets.
We have now clarified that:
“it is the first to use a large set of routinely collected data not limited to a clinical service in the UK.”

15 Lines 10-13 on page 11 – What makes number of procedures and diagnosis codes proxies for severity? Severity tells how urgently a patient has to be attended in the Emergency Department based of the presenting condition?
Thank you for pointing this out. We have revised the statement and have added the notion of comorbidity that these two proxies are relevant to.

16 Further clarifications needed on:

- a. It would be good to understand the percentage of time spent in outlier ward/s and percentage of time spent in home ward/s for both groups.
- b. Statistical analysis needs further work. For example presenting univariate results when multivariate modelling was done does not add value but is confusing.
- c. When analysing readmission – were the patients who died in-hospital removed from the analysis?
 - a. Percentage of time would be impossible to calculate as we did not have time stamps on the patients “ward movements”.
 - b. Statistical analysis have been changed.

P-value was added in table 1 showing differences between the two groups.

We believe it is good practise to present univariate results as it is the first sign of association between the dependent and independent variables. Of course the multivariable model is more informative as it adjusts for confounding factors but that should not limit the publication of the univariate analysis.

In this paper, univariate analysis was used to calculate p-values for descriptive analysis purposes and we do not make any claims of association based on this part of the results.

For the multivariable models we have reported the effect size/degree of association that medical

outliers have on the patient outcomes (see table 3) rather than present the entire results from the four different models as we wanted to emphasise the potential association between the two sets of variables. We would be happy to include details of the multivariable analysis if the editors felt the exposition of the results would be improved.

c. We performed the analysis for re-admission excluding patients that died in hospital. In light of this comment, we have modified the effect values for readmission and we are making it clear in the methods section that we excluded patients that died in hospital from the readmission analysis.

17 There are minor spelling and grammatical errors, which need attention.
We hope our revision has addressed these minor errors.

VERSION 2 – REVIEW

REVIEWER	Lua Perimal-Lewis Flinders University of South Australia, Australia
REVIEW RETURNED	28-Mar-2017

GENERAL COMMENTS	The reviewer completed the checklist but made no further comments.
--